# Functionalized Hydrogels for Cartilage Repair: The Value of Secretome-Instructive Signaling

**DOI:** 10.3390/ijms23116010

**Published:** 2022-05-27

**Authors:** María Julia Barisón, Rodrigo Nogoceke, Raphaella Josino, Cintia Delai da Silva Horinouchi, Bruna Hilzendeger Marcon, Alejandro Correa, Marco Augusto Stimamiglio, Anny Waloski Robert

**Affiliations:** Stem Cells Basic Biology Laboratory, Carlos Chagas Institute—FIOCRUZ/PR, Curitiba 81350-010, Brazil; mjbarison@gmail.com (M.J.B.); rodrigo_nogoceke@hotmail.com (R.N.); raphaellajosino3006@gmail.com (R.J.); cintiahorinouchi@yahoo.com (C.D.d.S.H.); bruna.marcon@fiocruz.br (B.H.M.); alejandro.correa@fiocruz.br (A.C.); marco.stimamiglio@fiocruz.br (M.A.S.)

**Keywords:** cartilage, secretome, hydrogel, growth factors, extracellular vesicles

## Abstract

Cartilage repair has been a challenge in the medical field for many years. Although treatments that alleviate pain and injury are available, none can effectively regenerate the cartilage. Currently, regenerative medicine and tissue engineering are among the developed strategies to treat cartilage injury. The use of stem cells, associated or not with scaffolds, has shown potential in cartilage regeneration. However, it is currently known that the effect of stem cells occurs mainly through the secretion of paracrine factors that act on local cells. In this review, we will address the use of the secretome—a set of bioactive factors (soluble factors and extracellular vesicles) secreted by the cells—of mesenchymal stem cells as a treatment for cartilage regeneration. We will also discuss methodologies for priming the secretome to enhance the chondroregenerative potential. In addition, considering the difficulty of delivering therapies to the injured cartilage site, we will address works that use hydrogels functionalized with growth factors and secretome components. We aim to show that secretome-functionalized hydrogels can be an exciting approach to cell-free cartilage repair therapy.

## 1. Introduction

Cartilage is a dense, load-bearing, and flexible type of supporting connective tissue. It is essential for proper bone growth and functions as structural support, resists deformation, and lubricates the joints. Cartilage is avascular, aliphatic, without innervation, with few cells (chondrocytes) and a complex extracellular matrix (ECM) network. The different cartilage types present in our body are distinguished not only by their location but also by the composition of their ECM. While the hyaline cartilage present in synovial joints, growth plates, and ribs, among others, is rich in type II collagen (COL2) and aggrecan (ACAN), the fibrous cartilage present in intervertebral discs has a more significant presence of type I collagen (COL1). Elastic cartilage, found in outer ears, larynx, and epiglottis, contains elastic fibers in addition to the presence of COL2 and proteoglycans [1,2].

Moreover, transduction of mechanical stimuli (i.e., mechanotransduction) in chondrocytes influences their maintenance, viability, gene expression, chondrogenic activity, and ECM deposition [3]. Under normal conditions, the few chondrocytes present in cartilage can maintain the composition of the ECM in a balance between degradation and production of ECM proteins. However, injuries caused by mechanical shocks or other diseases can disrupt this balance. A pro-inflammatory environment enhances the expression and activity of ECM degrading enzymes, such as matrix metalloproteinases (MMPs) and aggrecanases (ADAMTs, disintegrin-metalloproteinase with thrombospondin motifs), resulting in a higher ECM degradation/synthesis ratio impairing cartilage homeostasis and function [4,5].

Cartilage injuries usually cannot heal, and due to the lack of nerves, the detection of the problem usually occurs in advanced stages, e.g., osteoarthritis (OA). According to a systematic review carried out by Cui et al. (2020), around 650 million people worldwide suffered from knee OA in 2020, with a prevalence of 22.9% in the population over 40 years of age [6]. OA is a multifactorial whole joint disease characterized by progressive cartilage degeneration, leading to joint space reduction, osteophyte generation, inflammation, and even damage to the underlying bone [7]. A similar scenario is observed in the progression of intervertebral disc degeneration [8]. More details about these and other cartilage diseases can be found in [1,9].

The initial treatment for cartilage damage consists of anti-inflammatory drugs and physiotherapy to treat the symptoms. Depending on the pathology, the treatment can also include changing healthier habits. Non-invasive therapy options have also been investigated. For instance, several studies have shown the beneficial effect of using photobiomodulation, with low- or high-level laser therapies in in vivo models or in patients with OA [10,11,12]. The effects of these therapies included reduction of pain, improvement in cartilage function and thickness [13,14,15], stimulation of angiogenesis [16], improvement in the anti-inflammatory [17,18,19], and antioxidant [14] responses. However, in many cases, these treatments alone are not enough to restore cartilage, and surgical interventions become necessary. Therapies that aim to stimulate the patient’s bone marrow to mobilize cells to the injured area for joint repair are one of the most used. Through techniques such as abrasion chondroplasty, microfracture, or microdrilling, the cartilage is exposed to the bone marrow (BM). The bleeding allows the formation of a clot and interaction with BM stem cells. Other treatments include mosaicplasty, osteochondral allografting, autologous chondrocyte implantation, and particulate juvenile articular cartilage. These techniques are described in detail and with application examples in [9,20]. Despite the beneficial result obtained with these interventions, the tissue formed is more fibrous cartilage than the original hyaline cartilage, and the surgery may have secondary consequences. New approaches have focused on tissue engineering and regenerative medicine strategies, including using signaling molecules, e.g., platelet-rich plasma or growth factors (GFs), and cell therapy, which may or may not be associated with matrices and scaffolds [21,22,23,24,25]. The main treatment options for cartilage injury and the new approaches are represented in Figure 1.

Cell therapy is based mainly on the use of stem cells, as they present critical properties for tissue regeneration, such as self-renewal and differentiation potential for several cell lineages. Strategies for cartilage regeneration using induced pluripotent stem cells (iPSCs) or adult stem cells, primarily mesenchymal stem cells (MSCs), are described elsewhere and thoroughly reviewed here [26,27,28]. In the last decade, several works have associated the beneficial aspects of stem cell therapies with the factors secreted by them, which act via paracrine signaling and influence the cellular microenvironment [29,30,31,32]. The complete set of components secreted by cells, the conditioned medium (CM) or secretome, consists of a soluble fraction and a vesicular component. Among soluble active factors secreted by cells, there are cytokines, chemokines, interleukins (ILs), GFs, adhesion molecules, hormones, and nucleic acids, as micro-RNAs (miRNAs), long non-coding RNAs (lncRNAs) or messenger RNAs (mRNAs) [33,34]. The vesicular component comprises the extracellular vesicles (EVs), particles delimited by a lipid bilayer classified as a function of size and biogenesis. EVs also contain bioactive factors such as lipids, proteins, and miRNAs [35]. These bioactive factors have anti-inflammatory, immune-modulatory, and angiogenic properties that play a relevant role in the therapeutic potential of stem cells allowing the implementation of cell-free therapies in regenerative medicine, avoiding disadvantages such as the heterogeneity of the MSC population, potential rejection risk, low cell survival in the receiving patient, or non-specific accumulation of MSCs in different tissues, among others [33].

Considering cartilage characteristics, delivering drugs, molecules, cells, and others to the lesion site is a challenge. An alternative is the use of matrix scaffolds, mainly hydrogels. They are promising candidates for cartilage tissue engineering due to their properties and similarity with the ECM [36,37,38] and for having properties that allow local application [39,40]. Hydrogels are composed of cross-linked polymers that form solvent-saturated 3D networks, which allow size-selective diffusion of macromolecules, nutrients, and other solutes [41,42]. Hydrogels with different compositions and mechanical properties have been used both alone to stimulate migration and differentiation of resident stem/chondroprogenitor cells [39] and in combination with cells such as stem cells or chondrocytes [43,44] or GFs (discussed below). In addition to the potential for delivering molecules, hydrogels can also be tailored to enable the development of structures that mimic the mechanical properties of cartilaginous tissue [37].

This review is based on studies that use total secretome or EVs, primarily from stem cells, in cartilage regeneration. The obtention of bioderivatives with more significant chondroregenerative potential by priming the secretome content will also be discussed. In addition, considering the dense ECM and the difficulty of delivering treatments to the cartilage, the review will address the use of hydrogels to deliver GFs and the secretome itself. Our objective is to highlight the use of potentialized secretomes as chondroregenerative inducers by delivering them into ECM-mimetic hydrogels, an exciting and versatile strategy to foster cartilage repair.

## 2. Application of Secretomes for Cartilage Repair

Over the years, prospective studies have been carried out based on CM or EVs isolated from the cell secretome for cartilage repair. Here, we will discuss some of these studies describing the use of the secretome of stem cells derived from different origins in in vitro and in vivo experiments to evaluate how paracrine factors increase chondrocyte and cartilage regeneration.

Co-cultures methodologies were initially used to observe the influence of the paracrine signaling in chondrocytes stimulated by an inflammatory microenvironment. When chondrocytes are stimulated with the inflammatory cytokine tumor necrosis factor-α (TNF-α) and co-cultured with adipose-derived MSCs (ASCs), the paracrine factors secreted by these MSCs increase chondrocytes viability and decrease the expression of metalloproteinase (MMP)-13, suggesting that secretome of ASCs play a role in the chondroprotective effect observed in cell therapy [45]. In an experimental model of chondrocytes and synoviocytes isolated from OA patients undergoing total knee arthroplasty, co-culture with ASCs also induced a decrease in the expression by the target cells of all inflammatory factors evaluated. ASCs can sense an inflammatory environment and secrete paracrine factors to reduce inflammation, probably via the COX-2/PGE2 pathway [46].

The secretomes obtained from MSCs of different origins were tested on cartilage repair in vitro to better understand the effects of soluble factors. CM from ASCs increases the expression of COL2 and decreases MMP activity in the supernatant of OA chondrocytes stimulated with IL-1β. Furthermore, the expression levels of some matrix metalloproteinases (MMP-3 and MMP-13) and pro-inflammatory cytokines decreased while the expression of IL-10, an anti-inflammatory cytokine, increased [47]. In a different in vitro model, chondrocytes were stimulated with TNF-α to induce hypertrophy and treated with CM from ASCs. It was shown that the secretome treatment exerts an anti-hypertrophic role in blunting the MMP activity induced by TNF-α via TIMPs (tissue inhibitors of MMPs), factors present in ASC-CMs [48]. Platas and colleagues also showed that CM from ASCs protects OA chondrocytes stimulated with IL-1β from the senescence process induced by the inflammatory condition, decreasing oxidative stress [49].

In addition to ASC secretome, other MSC sources were also used to collect CM, inducing similar outcomes when its chondroprotective effect is evaluated in vitro. Chondrocytes stimulated with lipopolysaccharide (LPS) to induce an inflammatory response and treated with CM from BM-derived MSCs (BM-MSCs) exhibited a decrease in the expression of genes associated with inflammation and free-radicals (TNF-α, IL-1β, IL-6 and iNOS), and upregulation of ACAN, an integral part of the ECM in cartilaginous tissue, in a dose-dependent manner [50]. Comparable results were obtained when chondrocytes stimulated with IL-1β were treated with CM produced by MSCs derived from human exfoliated deciduous teeth (SHED): decreased MMP-13 and NF-kB expression and increased expression of ECM related genes as (ACAN) and COL2 [51].

As in many adult tissues, MSCs were also isolated from the synovial membrane (SM-MSCs) [52]. Indirect co-culture of these cells and costal chondrocytes (CCs) showed that CM produced by SM-MSCs induced proliferation of CCs and a decrease in type X collagen (COLX) expression, suggesting that paracrine factors from SM-MSCs could participate in the anti-hypertrophic effect induced by these cells [53].

The chondroprotective effect of MSCs-CM treatment observed in vitro was also validated in vivo using different animal models of cartilage repair. A study comparing the effect of ASC-CM and shock-wave (SW) treatment, a non-invasive approach that promotes cartilage regeneration in an OA rat model, showed that CM treatment restores the hyaline cartilage, reduces the expression of inflammatory factors, and increases the expression of cartilage related genes as SOX9 and COL2. This effect was dose-dependent and with an effectiveness similar to the one observed when SW was applied [54]. Intraarticular injection of BM-MSCs-CM in an animal model attenuates the progression of OA by increasing the ratio of TIMP1 to MMP-13 and decreasing apoptosis while enhancing autophagy of chondrocytes [55]. Using an antigen-induced model of arthritis (AIA), Kay and colleagues observed that treatment with CM from BM-MSCs reduced inflammation and cartilage damage in the arthritic knee joint. The authors showed that the immunosuppressive activity resulted from an increase in Treg cells function and improvement of the Treg: Th17 ratio [56]. Similarly, Ogasawara and colleagues applied CM from SHED-MSCs in a mechanical-stress-induced murine model of temporomandibular joint OA. In line with the results discussed above, CM induces condylar cartilage and subchondral bone repair, decreasing the expression of IL-1β, iNOS, and MMP-13 in OA chondrocytes and diminishing the number of apoptotic cells [57].

Results discussed here highlight the protective and regenerative effect of CM produced by MSC from several origins and applied using in vitro and in vivo strategies, focusing on different cartilage degenerative diseases. Paracrine factors present in CM can induce a decrease in the catabolic processes that occur in damaged cartilage, such as matrix degradation through MMPs and the production of pro-inflammatory cytokines. At the same time, CM enhances anabolic processes by increasing the expression of genes involved in ECM synthesis and induces an anti-inflammatory environment. As a result, CM restores the imbalance that results in cartilage degradation caused by the inflammatory scenario observed in patients suffering from degenerative cartilage diseases.

The characterization of MSC secretome composition has been described by several groups [58,59]. Its composition varies according to the cell source, culture medium, culture time, and others. However, several studies described soluble factors with regeneration potential. MSCs secrete several GFs involved in the cross-talk with chondrocytes, such as insulin-like GF-1 (IGF-1) [60], transforming GF (TGFs) [61], and bone morphogenetic protein-2 (BMP-2) [62], which participate in several processes related to chondrocyte anabolism and cartilage matrix repair. These factors will be discussed in association with hydrogels in Section 3.1. Besides GFs, other factors are secreted by MSCs. For example, thrombospondin-2 (TSP-2), identified in the secretome of umbilical cord blood-MSCs, could stimulate chondrogenic differentiation of chondroprogenitor cells and promote cartilage regeneration in vitro and in vivo [63]. Other factors secreted by MSCs and related to cartilage homeostasis are TIMPs. These paracrine factors are regulators of ECM metabolism and upregulated under inflammatory conditions [64], where an imbalance between synthesis and matrix degradation occurs. Furthermore, the hepatic GF (HGF) was identified in the secretome of MSCs as a key paracrine factor involved in the anti-fibrotic effect of ASCs when co-cultured with OA chondrocytes [65].

The MSC secretome also presented soluble factors with anti-inflammatory and immunomodulatory functions. TGF-β, prostaglandin E2 (PGE2), and TNF-inducible gene (TSG)-6 are some examples. The secretion of these factors can occur induced by other cytokines, mainly inflammatory, such as IL-1β, TNF-α, and IFNγ [66]. Thrombospondin 1 (TSP-1) was also identified as a chondroprotective factor that reduces inflammation [67]. Factors such as IL-6 [68] and IL-1RA [69] mediate the anti-inflammatory response of MSCs in vivo models of arthritis.

Recently, in a comparative proteomic analysis of the secretomes from ASCs, BM-MSCs, and SM-MSCs exposed to IL-1β and TNF-α was observed that the inflammatory stimulus enhances the production of paracrine factors involved in cartilage repair. For instance, MSCs secreted proteins related to the TGF-β signaling pathway, a relevant pathway involved in collagen and aggrecan expression [70]. Furthermore, TSG-6 and TSP-1 were secreted only under an inflammatory environment, suggesting a role in cartilage recovery [71].

Besides the effect of the soluble factors, another active component of the secretome are the EVs. They are particles delimited by a lipid bilayer, without replicative potential, containing factors such as nucleic acids (mRNAs, miRNAs, lncRNAs, etc.), proteins, lipids, and small molecules [35]. Based on the International Society for Extracellular Vesicles (ISEV), EVs can be classified based on their size (small, medium, or large), density (low, middle, high), surface markers, or cell origin/ culture condition [72]. Nevertheless, a commonly used nomenclature is exosomes (EXOs; 40–120 nm), produced by the endosomal pathway, and microvesicles (MVs; 50–1000 nm), which originated from the plasmatic membrane [35,73]. Several works demonstrated the protective effect of EXOs and MVs and their cargoes in chondrocytes and cartilage repair.

Tofiño-Vian and colleagues showed that EXOs and MVs presented in the CM of ASC counterbalance the effects of IL-1β induction in OA chondrocytes, replicating the effects observed when whole CM was applied as a therapeutic agent [74]. As part of the regenerative mechanism of these MVs, it was recently suggested that peroxiredoxin-6 could mediate the protection of EVs against the oxidative stress induced by IL-1β in OA chondrocytes [75]. EXOs derived from BM-MSCs increase viability and decrease mitochondrial-induced-apoptosis in OA-like chondrocytes with the involvement of p38, ERK, and Akt pathways [76]. Similarly, using in vitro and in vivo approaches, He et al. demonstrated that exosomes derived from BM-MSCs attenuate the inflammatory effects of IL-1β induction, as well as its effect on ECM degradation/synthesis ratio in vitro while improving the knee pain in the rats induced to OA [77].

Exosomes secreted by MSC derived from embryonic stem cells (iMSC) also present chondroprotective roles. Applied to osteochondral defects in rats, EXO isolated from iMSCs increased cell proliferation and reduced apoptosis while modulating inflammation by increasing the M2-macrophage population and decreasing IL-1β and TNF-α, cytokines secreted by M1 macrophages [78].

Different studies have described the role of the EVs cargoes in the regenerative activity of the EVs. Ruiz and colleagues showed that the chondroprotection exerted by the BM-MSC-EVs was due to the presence of TGFBI mRNA and protein [79]. Other authors indicated that the non-coding RNAs presented in EVs are responsible for the regenerative effects of EVs. Micro-RNAs act as gene expression regulators modulating several processes in the cell [80], balancing cartilage homeostasis/disease, and posing as biomarkers [81]. For example, miR-127-3p, enriched in exosomes derived from BM-MSC, targeted the expression of cadherin-11 (CDH11), a protein involved in joint inflammation [82], in OA-like chondrocytes in vitro. Additionally, the authors observed a blockage in the activation of the Wnt/β-catenin pathway, a signaling pathway involved in the modulation of arthritis, evidencing that miR-127-3p exerts a chondroprotective effect on OA-like chondrocytes [83].

Circular RNAs (circ-RNAs) are non-coding RNAs involved in gene expression regulation and modulation of degenerative diseases [84]. The circRNA_0001236 was found enriched in exosomes from MSC induced to chondrogenic differentiation by 21 days, suggesting a role in cartilage formation. Exosomes from MSC overexpressing circRNA_0001236 decreased cartilage degradation and slowed OA progression in a mouse model. CircRNA_0001236 contains a target site to the miR-3677-3p, a regulator of SOX9, which was upregulated by the circRNA. The authors proposed that circRNA_0001236 could act as a miR-3677-3p sponge, favoring the ECM anabolism/catabolism in the OA context [85].

Results presented here (summarized in Figure 2) and elsewhere [27,86] point to the relevance of the secretome of stem cells in cartilage regenerative medicine and its potential application as cell-free therapies. In addition, new strategies developed by conditioning stem cells to increase their regenerative capacity will be discussed in the next section.

### 2.1. Modulation of Secretomes

A recognized strategy to improve the regenerative properties of cell secretomes consists of culture cells under specific priming conditions, optimizing the functionality of the paracrine factors secreted by them. To that end, different approaches can be applied, such as modified physical conditions (reduced oxygen tension) [31,87,88], specific materials for cell culture [89], surface topography [90,91], the addition of molecules to the cell culture (cytokines or growth factors) [92,93] and cell growth as spheroids (3D cultures) [94,95]. For a detailed review of how MSC secretome can be modulated to enhance angiogenesis, immunomodulation, and other properties, see Miceli et al. (2021) [96]. Here, we will focus on several examples of priming conditions that increase the chondroprotective effects of the secretome, especially from MSC.

One of the most established methods to increase the regenerative properties of secretomes is the culture under hypoxic conditions. In vitro cell culture is routinely carried out in an atmosphere with 21% oxygen. Differently, in the human body, cells reside under lower oxygen tensions. For example, MSCs in adipose tissue and bone marrow live in a hypoxic microenvironment (to 1–9% oxygen) [97,98] as well as chondrocytes [99]. Both in vitro and in vivo experiments showed that CM from BM-MSCs grown under hypoxia increased chondrocyte proliferation and migration while decreasing apoptosis compared to secretome from cells grown in normoxia. Furthermore, hypoxia modified the expression and loading of miRNAs in EVs, increasing the expression of miRNAs involved in signaling pathways potentially related to chondrocyte survival [100].

The addition of growth factors to MSCs culture also increases the regenerative potential of its secretome. IGF-1 is an anabolic growth factor involved in cartilage homeostasis [101]. When pre-conditioned with IGF-1, SM-MSCs produced a secretome with higher chondroprotective properties, showing increased levels of growth factors such as BMP-2, FGF18, and TGF-β1 that induced a higher expression of COL2 and repressed the expression of MMP-13 and ADAMTS4, in an OA model [102]. Similarly, EXOs isolated from BM-MSCs pre-conditioned with TGF-β1 presented higher levels of miR-135b than the non-primed MSCs. EXOs enriched with miR-135b attenuated cartilage degradation by inducing the polarization of synovial macrophages to the M2 phenotype through the targeting of MAPK6 [103]. As previously discussed here, MSCs can sense an inflammatory environment and respond by different immunomodulatory mechanisms. When BM-MSCs were primed using inflammatory cytokines, such as TNF-α and IFNγ, the secretome was enriched with immunomodulatory factors such as IDO1 and IL-6. This modulated CM induced an anti-inflammatory response when applied to explants of cartilage and synovium and influenced the ECM turnover [104].

Another factor influencing the secretome composition is the culture system, such as tridimensional culture conditions or growing cells on specific materials. Wang and collaborators showed that the use of low doses of MSCs loaded on gelatin microcarriers as a therapeutic agent in a rat OA model presented a similar regenerative effect compared to repeated and high doses of MSCs. Investigation of this in vivo observation showed that supernatants from co-culture of chondrocytes and MSCs in 3D constructs had fewer inflammatory factors than co-culture with MSCs in monolayers. An increased expression of genes related to chondrogenesis and ECM interactions were also observed, suggesting a high therapeutic potential of secretome when MSCs are cultured in 3D constructs [105]. Similarly, when MSC was encapsulated in alginate microbeads and co-cultured with patient-derived OA chondrocytes, paracrine factors induced a pro-regenerative microenvironment, increasing anabolic, proliferative, and anti-apoptotic processes [106].

The use of specific materials and topography for culturing MSCs also influences the composition of its secretome. Kadir and collaborators obtained CM of MSCs cultivated on electrospun fibers composed of poly-L-lactide-co-ε-caprolactone (PLCL). Interestingly, the authors showed that this CM had enhanced regenerative properties in promoting chondrocyte proliferation and MSCs chondrogenesis, controlling inflammation, and increasing the anti-apoptotic activity compared to the CM obtained from traditional 2D cultures. Moreover, the use of aligned or randomly organized fibers affected the CM profile and the cellular responses demonstrating how the topography of the cell-substrate influences cell-ECM interaction via mechanotransduction signaling pathways [107]. The physical properties of the cell culture substrate and the resulting mechanotransduction responses in MSCs can fine-tune their secretomes, a topic of relevance in the current literature [108].

Besides strategies to potentialize secretomes by priming cells under specific culture conditions, cells can be genetically modified to produce more functional secretomes or EVs or even modify EVs after their isolation. For instance, EXOs from synovial fibroblasts that overexpress miR-126-3p, a miRNA downregulated in OA patient-derived synovial fluid exosomes, induce a decrease in chondrocytes inflammation in vitro and restrict cartilage degeneration in an in vivo model [109]. Engineered EXOs are also being proposed for cartilage repair, specifically, using EXOs as targeting vehicles. Micro-RNA-140 is expressed in articular chondrocytes and exerts a protective role for cartilage [110]. Liang and colleagues developed engineered exosomes that encapsulate miR-140 and are targeted to cartilage by the fusion of Lamp2b, a protein present on EXOs membrane, with CAP, a chondrocyte affinity peptide. This strategy allowed the delivery of the therapeutic miRNA to the damaged tissue with higher retention after injection, improving the progression of OA in a rat model [111].

## 3. Hydrogel and Cartilage Regeneration

To be used in tissue engineering, the hydrogels ought to be biocompatible, non-cytotoxic, proper for tissue integration, and viable for clinical application [25,39]. In addition, to be used as a cartilage substitute, these hydrogels have to reflect the unique properties of the cartilage (described earlier in Section 1) and be mechanically stable. It is also desirable that they support and enhance a complete regenerative process, which involves the activity of different components such as the immune system, stem/progenitor cell proliferation, differentiation, and ECM remodeling. The hydrogel should also allow the formation of non-hypertrophic functional cartilage tissue. The use of mechanical stimuli or the addition of elements with the potential to reduce hypertrophy, such as chondroitin sulfate, platelet lysate, 5-bromoindole-2-carboxylic acid (BICA) are some of the options [112,113]. Besides these properties, hydrogels need to be efficiently integrated into the host tissue in order for them to function properly. To this end, several strategies were developed to increase hydrogel adhesion in tissue engineering, mainly introducing chemical modifications to the hydrogel, and creating covalent, electrostatic, or supramolecular bonding, among others (reviewed by [114,115]). For example, hyaluronic acid (HA) hydrogels were modified, introducing aldehyde groups and methacrylate (AHAMA) [116] and o-nitrobenzyl (NB) groups [117] which interact with surrounding tissue increasing adhesive strength and promoting cartilage regeneration.

Ideally, the engineered hydrogel should be gradually substituted by native ECM, and the biodegradability rate should be compatible with the progress of the regeneration process [36]. Both the polymer backbone and the cross-linking network may target the degradation activity [118]. Crosslinkers sensitive to enzymatic activity, for instance, are helpful for controlling the biodegradability necessary for the regeneration process [25]. The different cross-linking densities of the hydrogels also influence the secretion of matrix molecules such as glycosaminoglycans (GAGs), collagen (II and VI), ACAN, link protein, and decorin, as well as MMPs [119]. Moreover, the choice of different cross-linking methods allows the control of the hydrogel structure and stability. While physical methods such as temperature and pH involve reversible structural processes, the chemical methods generate covalently bound networks [36].

The diffusion rate of solutes, enzymes, factors, and other elements in the hydrogel depends on the relationship between the size of the molecule/particle and the mesh size of the polymer network [118,120]. Thus, the hydrogel porosity, which is related to its mechanical and physicochemical properties (swelling ratio, degradation resistance), influences the degradability rate of the hydrogel [121,122], nutrient diffusion, transport of oxygen, removal of toxic components, as well as the cell proliferation, migration, and adhesion [3,41,121,122,123]. Therefore, by modulating the mesh size, swelling, and degradability rate, it is possible to produce hydrogels capable of stabilizing molecules and other bioactive factors and releasing them in a controlled manner, enhancing and tuning the regenerative process. Different strategies have been developed to tailor the hydrogel features to deliver immunomodulatory and pro-regenerative molecules and factors in a controlled manner, which have been extensively reviewed elsewhere [118,120,124]. Some examples of the association between hydrogels with GFs and EVs and their release times are summarized in Table 1 and will be discussed in the next sections.

**Table 1 ijms-23-06010-t001:** Delivery systems and in vitro release time of growth factors and extracellular vesicles associated with hydrogels with different compositions.

Hydrogel Composition	GF/EXO/EV	Delivery System *	Release Time	Ref.
Thiolated chitosan + carboxymethyl cellulose	TGF-β1	Scaffold	21 days	[125]
Sulfated carboxymethyl cellulose + carboxymethyl cellulose + gelatin	TGF-β1	Scaffold	30 days	[126]
Alginate-poly(acrylamide)	TGF-β3	Poly(lactide-co-glycolide) nanoparticle	60 days	[127]
Oligo (poly(ethylene glycol) fumarate)	TGF-β1	Gelatin microparticles	28 days	[128]
Thiolated gelatin + poly(ethylene glycol) diacrylate	IGF-1	Poly(ethylene adipate)/ heparin coacervates	21 days	[129]
Silk fibroin hydrogel	TGF-β1 and BMP-2	Chitosan nanoparticles (TGF-β1); Scaffold (BMP-2)	Up to 15 days (both)	[130]
Silk fibroin hydrogel	MGF and TGF-β3	Scaffold	28 days (both)	[131]
Aldehyde-functionalized chondroitin sulfate (OCS) + gelatin methacryloyl (GM)	EXO from BM-MSCs	Scaffold	14 days	[132]
dECM + gelatin methacrylate	EXO from BM-MSCs	Scaffold	14 days	[133]
dECM	EXO from ASCs	Scaffold	28 days	[134]
O-nitrobenzyl alcohol moieties modified hyaluronic acids (HA-NB) + gelatin	EXO from hiPSCs-MSC	Scaffold	14 days	[135]
Poloxamer-407 and 188 mixture	PRP-EXO	Scaffold	1 month	[136]
Poly(D,L-lactide)-b-poly(ethylene glycol)-b-poly(D,L-lactide) (PDLLA-PEG-PDLLA, PLEL)	Small EVs (circRNA3503) from Synovium MSCs	Scaffold	up to 35 days	[137]
Gelatin methacrylate (Gelma) + nanoclay	Small EV from hUMSC	Scaffold	31 days	[138]

* Delivery systems are differentiated into: (1) factors or extracellular vesicles only incorporated into the hydrogel (scaffold) or (2) factors initially encapsulated and then associated with the hydrogel. ASC: adipose-derived mesenchymal stem cell, BM-MSC: bone-marrow-derived mesenchymal stem cell, BMP-2: bone morphogenetic protein 2, dECM: decellularized extracellular matrix, EV: extracellular vesicles, EXO: exosome, GF: growth factor, hiPSC-MSC: human-induced pluripotent stem-cell-derived MSCs, hUMSC: human umbilical-cord-derived mesenchymal stem cell, IGF-1: insulin-like growth factor 1, MGF: mechano growth factor, PRP: platelet-rich plasma-derived, TGF-β: transforming growth factor β.

In diseases such as OA, the tissue damage induces an inflammatory process that leads to the secretion of catabolic enzymes and more tissue degeneration, eliciting a higher inflammatory response [139]. Controlling the inflammation and the immune system activity to allow a healthy transition to the next steps of tissue regeneration is essential to achieving proper chondrogenesis. Then, it is desirable to associate a hydrogel composition that uses biomaterials with immunomodulatory potential, such as xanthan gum, heparin/heparin sulfate, sulfated alginate, and carboxymethylated chitin, with the delivery of immunomodulatory molecules or drugs (corticosteroid and non-steroid immunomodulatory drugs), to control the timing of this process [140].

It is also noteworthy that the cartilage is a non-homogeneous tissue containing different layers with unique features [38]. The development of hydrogels must envision this complexity to achieve functional and physiological regeneration [38]. To this end, different types of hydrogels have been designed using synthetic, natural, or hybrid polymers that have been extensively reviewed by other authors [25,36,38,42]. In general, synthetic polymers such as polyethylene glycol (PEG), polycaprolactone (PCL), and polyvinyl alcohol (PVA), among others, have high mechanical strength and good reproducibility [25,38]. A downside is that they may have limited bioactivity; however, the features of these hydrogels can be tuned out by chemical and structural manipulation [36].

Natural polymers might be composed of peptides, proteins (e.g., collagen, gelatin, silk fibroin, sericin, elastin), and polysaccharides (e.g., hyaluronic acid, chitosan, alginate, chondroitin sulfate, agarose) [25,36,38]. Natural hydrogels tend to have greater biocompatibility and biodegradability than synthetic polymers, have a low immune response, and may even contain bioactive motifs [36]. The disadvantages are low mechanical resistance and degradability control, among others [25,36,140]. Another promising approach is using hydrogels composed of decellularized extracellular matrix (dECM), mainly from cartilage. These hydrogels may be obtained by removing the cells from the native tissue, followed by enzymatic digestion, neutralization, and temperature gelation [141]. Even though they also have a limitation concerning mechanical stability, dECM-based hydrogels have the advantage of being composed of natural matrix components, with improved biocompatibility and potential bioactivity [38,141]. The pros and cons of natural and synthetic hydrogels might be overcome by combining and tuning hybrid compositions to achieve better results [36]. Moreover, the hydrogel’s versatility may be potentialized by the association with factors and other bioactive elements with regenerative potential.

### 3.1. Growth-Factor-Functionalized Hydrogels for Cartilage Repair

A major challenge of tissue engineering is to recreate physical–chemical conditions similar to the physiological environment of cells, mimicking the structure of native tissue [118]. Cartilaginous tissue formation in vivo is driven by multiple signals that integrate in an orchestrated manner. Among these signals are GFs, polypeptides capable of inducing and promoting cascades of events that influence cell proliferation, differentiation, and migration [142]. A repertoire of growth factors such as BMPs, IGF, FGF, and TGF-β have been employed in cartilage tissue engineering. The most influential of these factors identified for articular cartilage development is TGF-β, which has a central role in regulating metabolic homeostasis and ensuring the structural integrity of cartilage tissue. In addition, it plays a critical role in tissue repair and regeneration [61]. A biologically correct representation includes attention to the dosage of GFs in the cell culture medium. It has already been shown that high doses of TGF-β1 might cause osteophyte development, preventing the establishment of a cartilaginous phenotype [143]. Furthermore, excess levels of IGF-1 decrease IGF-1 receptor mRNA [144] and COL2 deposition [145]. Therefore, strategies to control the release of these factors are needed to improve their regenerative capacity. Nevertheless, GFs are targets of proteolytic cleavage and often bind to molecules of the ECM, which can result in lower stability and a shorter half-life [146]. To circumvent these issues, incorporating GFs in hydrogels, directly, covalently bound, or in nanoparticles or microspheres has emerged as a promising approach [147,148,149,150].

Hydrogel scaffolds can protect GFs from degradation by entrapping the molecules in a matrix network, allowing a controlled release of the growth factors into the system [151]. In a recent study, Zhang et al. developed thiolated chitosan (TCS) and carboxymethyl cellulose (CMC) hydrogel, later embedded with TGF-β1. Treatment with TCS/CMC hydrogel associated with TGF-β1 in induced cartilage lesions in rat knees showed that standard hyaline cartilage was formed after eight weeks of transplantation, with cells encapsulated in gaps and arranged in columns [125]. CMC scaffold has also been used in its chemically sulfated form (sCMC), mimicking heparan sulfate and binding effectively and sequestering cationic growth factors such as TGF-β [152]. Consequently, this type of hydrogel has a long-term presentation of the factor. An in vitro experiment with MSCs encapsulated in the sCMC/gelatin hydrogel carrying TGF-β1 showed increased expression and deposition of hyaline cartilage markers and reduced fibrocartilage and expression of hypertrophy-related markers [126].

Saygili and colleagues (2021) used a hydrogel composed of a combination of polyacrylamide (PAAm) and alginate (Alg) [127]. In this system, TGF-β3 was released in a controlled manner through poly(lactide-co-glycolide) (PLGA) nanoparticles three-dimensionally embedded in the hydrogel. To test the effectiveness of the hydrogel in a model of chondral injury induced in rats, in situ applications of PAAm-Alg functionalized with TGF-β3 were performed. The results showed the formation of new tissue with a regular surface and an organized pattern of chondrocytes integrated into the surrounding tissue [127]. PLGA as a delivery platform of TGF-β3 was also used in association with other hydrogels, such as methoxy poly(ethylene glycol)-poly(alanine) (mPA) [153]. The incorporation of PLGA microspheres to carry TGF-β3 in the system showed a persistent release of GF for eight weeks. The controlled release of TGF-β3 enhanced the expression of cartilaginous markers such as COL2 and ACAN in cartilage cells isolated from rat knees while repressing the expression of the osteogenic marker COL1 [153].

The study of Park and collaborators (2007) investigated the potential of gelatin microspheres loaded with TGF-β1 and incorporated into oligo(poly(ethylene glycol) fumarate) (OPF) scaffolds to promote chondrogenic differentiation in rabbit MSCs [128]. They observed an increase in the expression of COL2 and ACAN markers in MSCs cultured in OPF with TGF-β1-loaded microspheres over the control cells that were not in contact with the GF [128]. This study showed that using gelatin microparticles to carry TGF-β1 could complement the OPF’s hydrogel ability to provide a good three-dimensional microenvironment [154,155] for chondrogenic differentiation of MSCs.

Along with TGF-β, IGF-1 is considered a promising bioactive molecule for articular cartilage repair by stimulating MSCs proliferation, expression of chondrogenic-related genes, and cartilage matrix formation [60]. In addition, IGF-1 suppresses MMP1, MMP3, IL-1, and TNF-α, which could prevent ECM degradation during cartilage repair [156]. Cho et al. encapsulated IGF-1 in poly(ethylene adipate)/heparin coacervates (spherical aggregates of lipid molecules held together by hydrophobic forces), which were later embedded in a gelatin (gelatin-SH)/poly(ethylene glycol) diacrylate (PEGDA) hydrogel. The coacervate hydrogel was able to maintain a slower release of IGF-1 compared to that from the hydrogel alone and improve the chondrogenic differentiation of ASCs in vitro [129]. The in vivo application of coacervate hydrogel in a rabbit knee osteochondral defect model showed that this combination promotes a better gross appearance and higher semi-quantitative histological scores in cartilage, indicating that the sustained release of IGF-1 improves in vivo osteochondral tissue regeneration [129].

The interaction between different GFs is an interesting strategy to maximize the healing potential of a hydrogel. BMP-2 can initiate chondrogenesis of MSCs [62] and stimulate the Smad2/3 signaling through the BMP receptor ALK3 [157]. Once this pathway is activated, cells can produce their own TGF-β, which will act through autocrine signaling [158]. While this makes BMP-2 an attractive bioactive molecule for cartilage repair, BMP-2 is also involved in hypertrophy differentiation [159] and endochondral ossification through the activation of the Smad1/5/8 signaling [160]. However, the presence of TGF-β3 could favor the Smad2/3 signaling over Smad1/5/8, which could suppress and delay the hypertrophic differentiation stimulated by the presence of BMP-2 [161]. Therefore, the combination of BMP-2 and TGF-β3 is an interesting approach to developing bioactive hydrogels. Along this line, Li et al. (2021) developed a silk fibroin (SF) hydrogel incorporated with chitosan (CS) nanoparticles carrying TFG-β1 and BMP-2, which were able to maintain a steady release of the GFs in the media. While the presence of the CS/SF hydrogels carrying each of the GFs individually was sufficient to improve tissue regeneration in a rabbit knee defect model compared to the control group, the combination of TGF-β1/BMP-2 offered better histological scores than any of the other groups evaluated [130].

Another study demonstrated that one isoform of IGF-1, the mechano growth factor (MGF), acted synergistically with TGF-β3 when both were embedded in SF scaffolds [131]. While MGF is known to increase the migration of MSCs [162], its influence on the chondrogenic differentiation of these cells is still unknown. In this study, MGF alone had no effects on chondrogenesis of human MSCs; however, it acted synergistically with TGF-β3, increasing cell recruitment by up to 1.8 and 2 times higher than the TGF-β3 group alone [131]. The presence of MGF also increased the TGF-β3-induced chondrogenesis of human MSCs and downregulated COL1 secretion, suggesting that this approach could prevent fibrocartilage formation during cartilage repair and regeneration [131].

It is remarkable that depending on the hydrogel composition and delivery system, the GFs or extracellular vesicles are released for 2 weeks reaching up to 60 days (Table 1). Collectively, these studies showed the advantages of using hydrogel-embedded GFs and microparticles as GF carriers to boost hydrogel’s ability to promote a chondrogenic inductive milieu.

### 3.2. Secretome-Functionalized Hydrogels for Cartilage Repair

Several hydrogels, composed of the most diverse materials, have been extensively used as scaffolds for 3D cell culture. This type of application has also been used to improve the quality and facilitate the obtention of the cell secretome, as addressed above (Section 2.1. Modulation of Secretomes). However, few studies addressed the use of hydrogels in combination with secretome as an acellular optimized complex to be considered a potential therapeutic tool for cartilage regeneration, although some examples can be found in the literature for other tissue regeneration models [32,163]. In addition, none of these few studies have used the entire secretome but the EVs present in them. The EVs were mostly isolated from MSCs from various sources, e.g., bone marrow, adipose tissue, umbilical cord, and others, combined with different types of hydrogels [132,133,134,135,137,138,164,165].

The main reason for loading the secretome into a hydrogel, similar to what was discussed for GFs, is to prevent the immediate washout of the active molecules and prolong therapeutic exposure seeking better effectiveness on cartilage regeneration. Many hydrogels are composed of synthetic materials which are easily handled and adjusted, present more stable biochemical properties, and are less susceptible to batch-to-batch differences [136,137]. A recent study has demonstrated that a mixture of Poloxamer-407 and 188 is a functional injectable carrier to deliver platelet-rich plasma-derived exosomes (PRP-Exo) in an OA model. The hydrogel could sustain the release of EXOs for about a month in vitro, and by using an imaging system, the authors confirmed the increase of PRP-Exo retention in the mice’s injured joint when the gel was applied. PRP-EXOs released from the hydrogel-PRP-EXOs complex remained functional, demonstrating positive effects on BM-MSC and chondrocytes in vitro and protective activity against inflammation-mediated apoptosis and degeneration of chondrocytes in vivo [136]. Another synthetic hydrogel potentially applied as a sustained-release delivery system was investigated by Tao and colleagues (2021) in OA models. Poly(D, L-lactide)-b-poly(ethylene glycol)-b-poly(D, L-lactide) (PDLLA-PEG-PDLLA, PLEL) triblock copolymer gels successfully incorporate and slowly released EVs derived from synovium MSC overexpressing the circRNA3503. The hydrogel complex with circRNA3503-loaded EVs protected the cartilage and delayed the progression of osteoarthritis when evaluated in vitro and in vivo [137].

Although synthetic compounds may be an option for hydrogel fabrication, most studies evaluating hydrogels in combination with secretome for cartilage repair have used biomaterials such as gelatin and chondroitin sulfate, which provide ECM components to the microenvironment that can facilitate the regeneration process [132,135,138,165]. Two studies have evaluated the delivery of EXOs derived from BM-MSCs using chondroitin sulfate-based hydrogels for cartilage regeneration approaches. First, Zhang and colleagues (2021) fabricated a highly adhesive hydrogel containing a cross-linked network of alginate-dopamine, chondroitin sulfate, and regenerated silk fibroin to improve adhesion to wet tissues. Exosomes encapsulated by this hydrogel promoted migration of BM-MSCs, differentiation into chondrocytes, and helped cartilage repair when applied in the rat patellar grooves model [165]. More recently, Guan and collaborators (2022) loaded EXOs from BM-MSCs in an ECM-mimic hydrogel containing aldehyde-functionalized chondroitin sulfate and gelatin methacryloyl seeking ECM synthesis and inflammation inhibition in a rat growth plate injury model. The efficacy of the complex was extensively explored through in vitro and in vivo experiments. The anti-inflammatory effect was attributed to the regulation of the polarization of M2 macrophages, which probably favored the anabolic activity of damaged chondrocytes and inhibited bony repair [132]. Shi and colleagues (2020) also observed M2 macrophage polarization as the mechanism modulating tendon-bone healing induced by BM-MSCs-derived EXOs embedded in a chitosan/β-glycerophosphate/collagen-based hydrogel [164].

Hyaluronic acid-derived biomaterials are also excellent candidates to be applied in the composition of hydrogels. A hydrogel tissue patch, fabricated from o-nitrobenzyl alcohol moieties modified hyaluronic acids (NB-HA) plus gelatin, complexed with EXOs isolated from human induced pluripotent stem cell-derived MSCs showed potential for cartilage regeneration. The NB-HA hydrogel allows an efficient hydrogel adhesion and tissue integration, while EXOs improve the viability, proliferation, and migration of BM-MSC and chondrocytes in vitro and induce cartilage repair in a rabbit articular cartilage defect model [135]. A limitation of using biomaterials such as gelatin is their poor strength. To overcome this, the use of nanomaterials may improve the mechanical, as well as the biological, characteristics of a hydrogel. The addition of nanoclay in a gelatin methacrylate hydrogel produced a potential material for cartilage regeneration which promoted the sustained release of EVs from umbilical cord MSC. Once released, the EVs promoted cartilage regeneration in a rat cartilage defect model. Authors suggest that these EVs are abundant in miR-23a-3p, which activates the PTEN/AKT signaling pathway promoting cartilage regeneration [138].

Other studies bring minimally processed natural materials such as decellularized ECM as the base for the composition of the hydrogels and show promising results in association with exosomes. The abundance of peptides in the ECM facilitates cell recruitment, cell infiltration, and differentiation. Furthermore, the high biocompatibility of ECM-hydrogels demonstrates advantages in relation to other biomaterials [133,134]. Chen and colleagues (2019) applied a decellularized cartilage ECM associated with gelatin methacrylate (GelMA) to biofabricate an ECM/GelMA/EXO complex using exosome isolated from BM-MSCs. The complex applied as a bioink for 3D printing scaffolds with radially oriented architecture has retained the EXOs for 14 days in vitro and seven days in vivo. Most importantly, exosomes enhanced the positive effect of ECM on chondrocytes migration in vitro and showed the best therapeutic activity promoting cartilage regeneration in a rabbit osteochondral defect model. The authors suggest that the presence of the ECM as natural material in the hydrogel increased the ability of sustained the release of exosomes from the complex [133].

In another context, decellularized ECM from the nucleus pulposus was used as a hydrogel in combination with exosomes derived from ASCs, and the complex (dECM@exo) was challenged as a treatment for intervertebral disc degeneration [134]. In this case, no extra materials other than ECM and EXOs were added to the complex to fabricate the scaffolds, and the sustained release of the exosome was observed over 28 days in vitro. Besides, dECM@exo showed a protective response in the intervertebral disc microenvironment in a rat model for disc degeneration. The effect was likely due to a synergic activity of dECM and exosomes, which modulate inflammatory complexes and metalloproteinases [134]. A summary of types of hydrogels, associated EXOs/EVs and their time of release can be found in Table 1.

## 4. Challenges and Perspectives

Over the years, many strategies have been developed trying to regenerate cartilage to its pre-injury condition. However, many challenges remain in the field, and the search for alternative therapies continues. Here we have shown how secretomes or EVs derived from stem cells have the potential for cartilage regeneration and how hydrogels can be the vehicle for their delivery.

In addition to the previously mentioned challenges in the treatment of cartilage lesions, which include the difficulty of treating an avascular tissue, with little innervation and few cells, the size of the lesion must also be considered. While initial injuries can affect small regions of the cartilage, there are other situations in which the subchondral bone can be damaged. Further, the continuous and often exacerbated inflammatory response that leads to cartilage ECM degradation is a point to be overcome. Thus, the development of new therapies should consider at which stage of the disease can be most effective, in addition to the possibility of developing specific treatments for each stage.

In this context, why use the secretome? The set of factors secreted by cells results from how cells sense the microenvironment in which they are inserted and how they respond to it, communicating with other cells. Cells can produce a wide range of factors adapted to a given situation that can act in various pathways, potentiating the effects, e.g., higher anti-inflammatory or proliferative potential.

However, many questions remain: what is the best cell source or best methodology for secretome collection? Should total secretome be used or subfractions as EVs are sufficient? Probably, different secretomes would be useful to treat different diseases or health problems. Thus, the first critical point is to identify the best cell source and culture conditions for these cells to secrete molecules with greater chondroregenerative potential.

Considering only the diversity of factors, many could be produced and combined individually. However, deciding which, which concentration, and how many factors to combine and producing each factor separately can increase the cost of therapy. Considering this, cells in culture could serve as a “factory” that, when priming for a specific response, generate a set of factors in quantities and composition necessary to treat a specific disease, reducing costs in the long term.

The vast majority of the reports indicate that MSC secretomes have beneficial effects, although it is not known precisely which molecules are responsible for the results observed. Advances in understanding the action of EVs in cells have indicated that mRNA, miRNAs, and proteins can help control the regenerative processes. Thus, when choosing the “ideal secretome,” it is desirable to characterize it extensively (e.g., mass spectrometry, antibody array, enzyme-linked immunosorbent assay, RNA sequencing), evaluate different fractions/molecules to identify what generates the effects, and obtain reproducible results.

The choice of the appropriate hydrogel composition for secretome delivery is another challenge. While hydrogels of many compositions have already been applied alone or with cells, the association with secretome or EVs for cartilage regeneration has been addressed more recently. The results obtained so far show the benefits of the synergistic use of these two components, associating the mechanical and drug delivery properties of the hydrogel with the therapeutic potential of the secretome. Furthermore, although some articles provide information on the release time of factors and secretomes trapped in hydrogels, few perform this verification in vivo. Moreover, considering the altered microenvironment at the injury site, there may be significant differences in the release of factors in vitro and in vivo.

The challenge is currently developing a hydrogel that serves as a temporary substitute for cartilage and allows the orchestrated release of the modulatory and stimulating factors of the secretome to promote the regenerative process, being gradually replaced by new cartilage tissue. In this sense, using hydrogels with complex compositions would be appealing, joining elements responsive to different phases of the regenerative process, capable of releasing the factors necessary for each of these steps.

Despite the challenges, functionalized hydrogels with a chondroregenerative secretome are a promising therapeutic option for cartilage regeneration (Figure 3).

## 5. Conclusions

In this work, we reviewed several pieces of work reporting the potential application of secretomes and their association with hydrogels in cartilage repair.

As described here, an ideal secretome (total or only EVs) for cartilage regeneration would be able to modulate the immune response and reduce inflammation at the lesion site, decrease levels of MMPs and increase the activation of resident chondrocytes to produce new ECM. Associated with this “ideal secretome,” a biocompatible and biodegradable hydrogel would integrate efficiently into the surrounding tissue, delivering the secretome in a controlled manner, and optimizing its function. The knowledge accumulated in the last years, together with the advances in, e.g., cell culture isolation and characterization of secretomes and hydrogel production have shown that there is still a need to improve the association between secretomes and hydrogels, which will bring opportunities for more efficient treatments in cartilage regeneration.

## Figures and Tables

**Figure 1 ijms-23-06010-f001:**
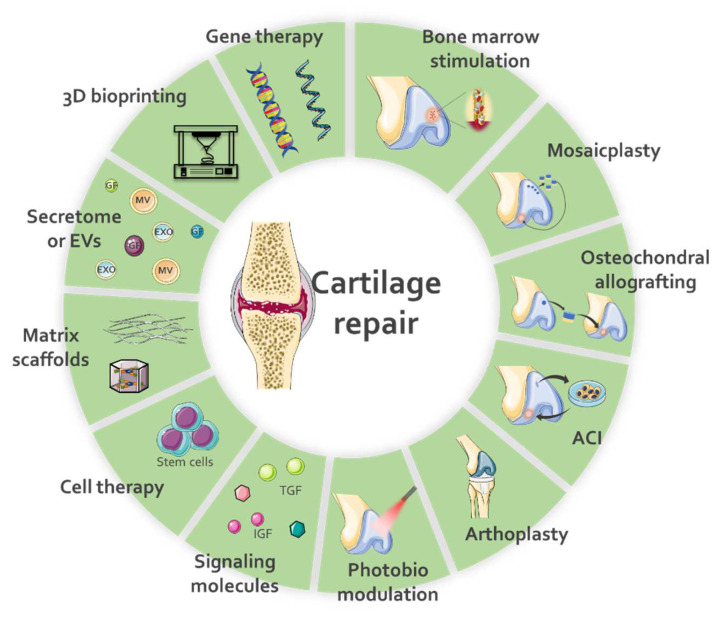
Cartilage repair treatments. This scheme represents some of the main treatments already used or under development for cartilage repair. ACI = autologous chondrocyte implantation; EVs = extracellular vesicles. The images were obtained from Servier Medical Art (http://smart.servier.com/, Accessed on 7 April 2022), licensed under a Creative Commons Attribution 3.0 Unported License.

**Figure 2 ijms-23-06010-f002:**
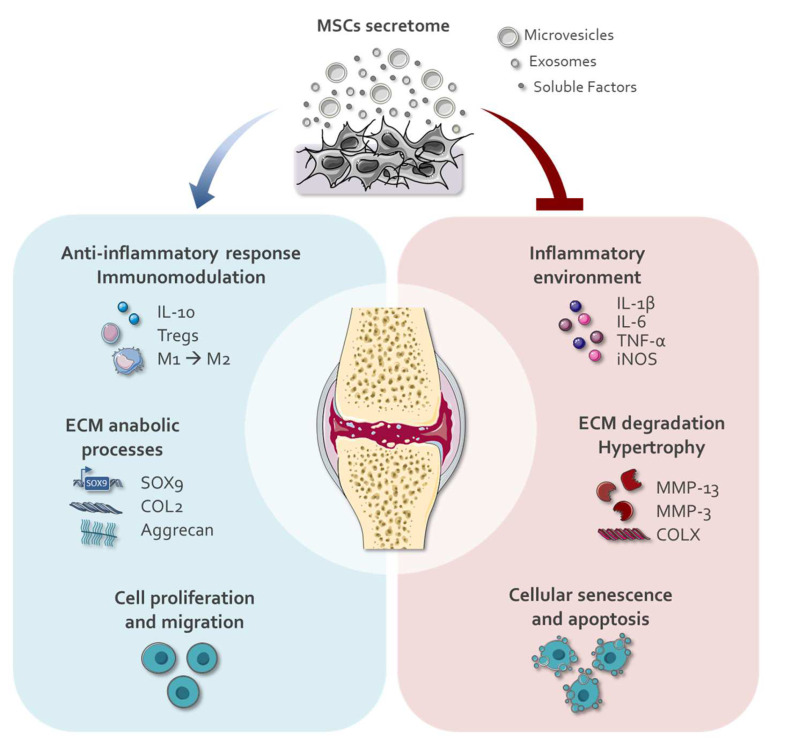
Effects of MSC secretome treatment on cartilage repair. Paracrine factors present in the MSCs secretome as soluble factors or contained in EVs (microvesicles and exosomes) can respond to cartilage injury. By inducing a decrease in the production of pro-inflammatory cytokines, a down-regulation of the matrix degradation through MMPs, and a reduction in cellular senescence and apoptosis, the MSCs secretome attenuates catabolic events at damaged cartilage. At the same time, the MSCs secretome favors cell proliferation and migration of chondrocytes, enhancing the expression of genes involved in ECM synthesis and inducing an anti-inflammatory response. The images were obtained from Servier Medical Art (http://smart.servier.com/, Accessed on 7 April 2022), licensed under a Creative Commons Attribution 3.0 Unported License.

**Figure 3 ijms-23-06010-f003:**
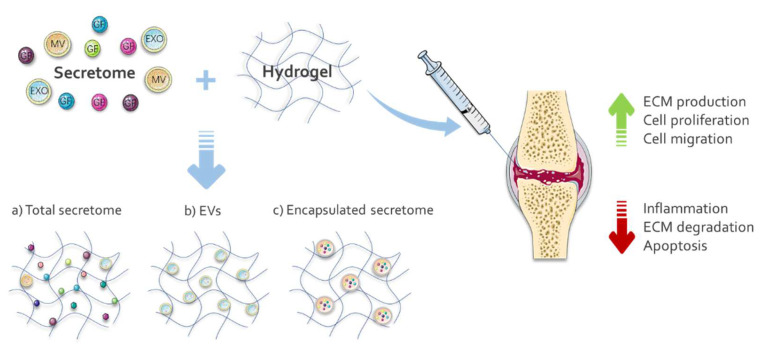
Functionalization strategies of hydrogels with secretome (total or only EVs) for treatment of cartilage damage. Secretomes, composed of soluble factors, as growth factors (GF), and extracellular vesicles (EVs) (microvesicles (MVs) and exosomes (EXO)), can be directly associated with hydrogel (with crosslinkers or not, (**a**,**b**)) or to be initially encapsulated and then incorporated with the hydrogel (**c**). The images were obtained from Servier Medical Art (http://smart.servier.com/, Accessed on 7 April 2022), licensed under a Creative Commons Attribution 3.0 Unported License.

## Data Availability

Not applicable.

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
