# Peer review of "Functionalized Hydrogels for Cartilage Repair: The Value of Secretome-Instructive Signaling"

_ijms, 2022, doi:10.3390/ijms23116010_

Round 1
Reviewer 1 Report
The manuscrip "Functionalized hydrogels for cartilage repair: the value of secretome-instructive signaling" by Barisón et al. is a well-written, useful summary of the possible methods and materials that can be used for the repair of cartilage tissue. As the title suggests it is mainly focusing on the use of growth factors, the secretome of different cell types, and the application of hydrogels for the delivery of those. The authors show several interesting and relevant results on the use of the aforementioned "materials" and their combination as well, and based on these, appropriately concludes their relevance for the promotion of cartilage healing.
I only have two minor notes:
- I think the legend for Fig 1. is too long, as the details of bone marrow stimulation and possible surgical techniques rather belong (and partly are) in the main text.
- In line 263, the correct is rather "was found", not founded.

Author Response
Response to Reviewer 1 Comments
The manuscrip "Functionalized hydrogels for cartilage repair: the value of secretome-instructive signaling" by Barisón et al. is a well-written, useful summary of the possible methods and materials that can be used for the repair of cartilage tissue. As the title suggests it is mainly focusing on the use of growth factors, the secretome of different cell types, and the application of hydrogels for the delivery of those. The authors show several interesting and relevant results on the use of the aforementioned "materials" and their combination as well, and based on these, appropriately concludes their relevance for the promotion of cartilage healing.
I only have two minor notes:
Point 1: I think the legend for Fig 1. is too long, as the details of bone marrow stimulation and possible surgical techniques rather belong (and partly are) in the main text.
Point 2: In line 263, the correct is rather "was found", not founded
Response for points 1 and 2: Thank you for your suggestions. We reduced the Fig1. legend and correct the phrase in line 263.

Reviewer 2 Report
The review is devoted to the cartilage repair strategies based on the application of cell secretomes and functionalized hydrogels. The perspective of different secretome formulations are discussed in view of their origin, cultivation regime, modification and method of delivery. The development of functionalized hydrogels and polymer scaffolds carrying the biologically active components, such as growth factors, peptides, signaling molecules, extracellular vesicles, etc., is considered in detail. Several strategies of targeted delivery of the treatment to cartilaginous tissue are outlined as well. The bibliography contains 150 citations with the most of them published in the recent decade.
In general, the review summarizes the new approaches of bioactive attenuation of cartilage degradation in osteoarthritis towards the stimulation of regenerative tissue response, ECM production, decrease of proteolytic activity and inflammation. The following issues should be considered prior to publication:
- In the Introduction section the common approaches for cartilage repair are given. However, the laser-based techniques of regeneration stimulation are missing. These methods should be included and discussed as well.
- Adhesion and diffusion properties of a particular formulation have critical importance for the final therapeutic effect. From the presented literature analysis, however, it is not clear which type of functionalized material is beneficial in terms of its adhesion and diffusion. Moreover, the observed therapeutic effects may originate not only from the bioactivity of a gel/secretome formulation, but also represent a cumulative effect of delivery effectiveness (permeation) and biological action. Thus, there is a need to differentiate the types of the discussed formulas according to their stability, adhesion and diffusion, especially when being used in vivo.
- Can the observed strategies be applied to all grades of osteoarthritis? Are there any recommendations for the concrete stage of the tissue degradation? How does the size and depth of a lesion influence the gel adhesion and the effectiveness of component diffusion?
- In the last section, containing the challenges and perspectives, the more critical analysis of the discussed strategies is needed. The most conclusions are connected with empirical findings, not with fundamental phenomena. Therefore, it is not quite clear what is the author’s position related to the future development of the field. The concrete scientific problems and directions to solve them should be mentioned.
Author Response
Response to Reviewer 2 Comments
The review is devoted to the cartilage repair strategies based on the application of cell secretomes and functionalized hydrogels. The perspective of different secretome formulations are discussed in view of their origin, cultivation regime, modification and method of delivery. The development of functionalized hydrogels and polymer scaffolds carrying the biologically active components, such as growth factors, peptides, signaling molecules, extracellular vesicles, etc., is considered in detail. Several strategies of targeted delivery of the treatment to cartilaginous tissue are outlined as well. The bibliography contains 150 citations with the most of them published in the recent decade.
In general, the review summarizes the new approaches of bioactive attenuation of cartilage degradation in osteoarthritis towards the stimulation of regenerative tissue response, ECM production, decrease of proteolytic activity and inflammation. The following issues should be considered prior to publication:
Point 1: In the Introduction section the common approaches for cartilage repair are given. However, the laser-based techniques of regeneration stimulation are missing. These methods should be included and discussed as well.
Response: We thank the reviewer for this suggestion. We included the laser-based technique for cartilage regeneration in the Introduction section, lines 58-63, highlighting the beneficial effects described for this therapeutic strategy. Also, we added a representative image in Fig 1.
Point 2: Adhesion and diffusion properties of a particular formulation have critical importance for the final therapeutic effect. From the presented literature analysis, however, it is not clear which type of functionalized material is beneficial in terms of its adhesion and diffusion. Moreover, the observed therapeutic effects may originate not only from the bioactivity of a gel/secretome formulation, but also represent a cumulative effect of delivery effectiveness (permeation) and biological action. Thus, there is a need to differentiate the types of the discussed formulas according to their stability, adhesion and diffusion, especially when being used in vivo.
Response: Thank you for your comments and suggestions. Currently, there is no consensus on which would be the best material for producing the hydrogel or the best way to associate a molecule with its composition. Throughout the articles discussed in the manuscript, we noticed that few of them showed information about the release time of drugs or factors, or even the degradation time of hydrogels. In addition, the studies that brought this information verified these parameters in vitro, with little work being performed in vivo. Our objective with this review was to highlight the potential use of the secretome as a cell-free therapy to regenerate injured cartilage, and that hydrogels could be used as delivery vehicles of the secretome. Considering the complexity of the subject and the great variation in the composition of hydrogels used to treat cartilage, we approach the topic more widely, showing the main characteristics of hydrogels to be used for cartilage repair and as delivery systems.
However, considering the importance of this topic highlighted by the reviewer, we have included in section “3. Hydrogel and cartilage regeneration” a sentence indicating some aspects of hydrogels that can modulate the diffusion of molecules (lines 395-411). Regarding the adhesiveness of hydrogels, we included some information in section “3. Hydrogel and cartilage regeneration” (lines 376-383) and some examples were presented in section “3.2. Secretome-functionalized hydrogels for cartilage repair” (lines 589-592; 608-609). Regarding the release time of factors or secretomes incorporated in hydrogels, to make the information clearer, we have included a table (Table 1) indicating hydrogel composition, which factor/secretome was used, how these molecules were associated with the hydrogel and the release time. It was possible to notice the variability of results and how few studies bring this information.
Point 3: Can the observed strategies be applied to all grades of osteoarthritis? Are there any recommendations for the concrete stage of the tissue degradation? How does the size and depth of a lesion influence the gel adhesion and the effectiveness of component diffusion?
Response: We appreciate and understand the reviewer’s concerns. The issues raised by the reviewer are really challenging and not addressed in depth in the current state of the art. We believe, when considering the importance of the inflammatory response in tissue repair, that the immunomodulatory potential of secretome itself may play an important role for all grades of osteoarthritis. Nonetheless, based on the different levels of cartilage injury, possibly different therapeutic strategies will need to be used. For example, larger lesions, in which the bone was also damaged, would require scaffolds and growth factors capable of regenerating not only cartilage but also bone tissue. Furthermore, different tissues (bone versus cartilage) have different biomechanical characteristics, which would affect the adhesion of a hydrogel. In this context, recently, biphasic hydrogel strategies have been developed, which have shown promise in the treatment of osteochondral lesions, promoting cartilage and bone regeneration. In addition, the entire microenvironment of the injured site can interact and react to the hydrogel in different ways.
These questions are also very interesting for us, but our aim in this review was to present a therapeutic possibility involving the use of secretome and hydrogel (as a delivery vehicle). We focused on showing examples of studies that use stem cell secretome, with or without association with hydrogels, in the treatment of cartilage lesions, avoiding studies that worked with osteochondral lesions.
However, considering the great influence of immune response in cartilage repair, we include one paragraph in section 3. (“Hydrogel and cartilage regeneration”, lines 412-420), showing examples of hydrogels with immunomodulatory potential and their association with immunomodulatory drugs.
Point 4: In the last section, containing the challenges and perspectives, the more critical analysis of the discussed strategies is needed. The most conclusions are connected with empirical findings, not with fundamental phenomena. Therefore, it is not quite clear what is the author’s position related to the future development of the field. The concrete scientific problems and directions to solve them should be mentioned.
Response: Thank you very much for the comment and for helping to make our review clearer. We revised the “Challenges and Perspectives” section, highlighting some of the current challenges in the use of secretomes and hydrogels for cartilage lesion therapies; and created a “Conclusion” section, with closing remarks.
